# Production and Quality of West Indian Cherry (*Malpighia emarginata* D. C.) under Salt Stress and NPK Combinations

Antonio Manoel da Silva Filho [1,*], Hans Raj Gheyi [1], Alberto Soares de Melo [2], André Alisson Rodrigues da Silva [1], Semako Ibrahim Bonou [1], Lumara Tatiely Santos Amadeu [11], Rener Luciano de Souza Ferraz [3], Patrícia Silva Costa [1], Lucia Helena Garofalo Chaves [1] and Rossana Maria Feitosa de Figueirêdo [1]

[1] Post Graduate Program in Agricultural Engineering, Federal University of Campina Grande, Campina Grande 58429-900, PB, Brazil; hans@pq.cnpq.br (H.R.G.); andre.alisson@estudante.ufcg.edu.br (A.A.R.d.S.); semako.ibrahim@estudante.ufcg.edu.br (S.I.B.); lumaratatielyea@gmail.com (L.T.S.A.); patriciagroambiental@gmail.com (P.S.C.); lucia.garofalo@pesquisador.cnpq.br (L.H.G.C.); rossana.maria@professor.ufcg.edu.br (R.M.F.d.F.)

[2] Department of Biology, State University of Paraiba, Campina Grande 58429-500, PB, Brazil; alberto.melo@servidor.uepb.edu.br

[3] Academic Unit of Development Technology, Federal University of Campina Grande, Sumé 58540-000, PB, Brazil; rener.luciano@professor.ufcg.edu.br

* Correspondence: manoel.silva@estudante.ufcg.edu.br

**Abstract:** This study aimed to evaluate the effect of fertilization combinations of nitrogen (N), phosphorus (P), and potassium (K) on the production and quality of West Indian cherry grown under salt stress in the second year of production. The study was conducted in a protected environment following a randomized block design with treatments distributed in a $2 \times 10$ factorial arrangement referring to two levels of electrical conductivity of irrigation water (0.6 and 4.0 dS m$^{-1}$) and 10 NPK fertilization combinations (80-100-100; 100-100-100; 120-100-100; 140-100-100; 100-80-100; 100-120-100; 100-140-100; 100-100-80; 100-100-120 and 100-100-140% of the recommendation in the second year of production), with three replicates and one plant per lysimeter. Production and post-harvest variables evaluated were: the total fruit weight, total number of fruits, mean fruit weight, the polar and equatorial diameter, total soluble solids, pulp pH, titratable acidity, maturity ratio, vitamin C, reducing sugars, total phenolic compounds, total anthocyanins, and flavonoids. The results indicate that irrigation with water having a salinity of 4.0 dS m$^{-1}$ negatively affected all production variables. The interaction between the ECw of 0.6 dS m$^{-1}$ and the 100-80-120 NPK fertilization combination increased the total number of fruits and the total fruit weight of West Indian cherry.

**Keywords:** post-harvest; flavonoids; anthocyanins; water salinity; fertilization management

## 1. Introduction

West Indian cherry (*Malpighia emarginata* D. C.) is a tropical fruit species of the family Malpighiaceae, having originated in the Caribbean islands and Central and South Americas [1]. The fruits of this species show high levels of ascorbic acid (vitamin C), phenolic compounds (benzoic acid, flavonoids, and anthocyanins), and total carotenoids [2]. Furthermore, West Indian cherry also stands out due its high nutritional and economic potential, as the fruit can be consumed fresh or processed into juices, ice creams, jellies, sweets, and other food products [3].

In northeastern Brazil, West Indian cherry cultivation has proven to be an economically important activity for the region, especially in its semi-arid portion, due to the species' adaptation to the local edaphoclimatic conditions [4,5]. However, it is known that the semi-arid region of Brazil is characterized, among other factors, by frequent water restrictions imposed by drought events, high evapotranspiration rates, and groundwater sources with

high salt concentrations [6]. Therefore, the quantity and quality of available water in the semi-arid region of Brazil has limited the large-scale production of West Indian cherry and other fruit crops in that region [7].

In this scenario, the use of water with high electrical conductivity (saline water) has become an increasingly common practice to irrigate crops such as West Indian cherry [7–9], passion fruit [10,11], and sugar apple [12,13]. However, high salt concentrations in water can compromise the growth, physiology, production, and post-harvest quality of fruits [13]. These disturbances occur mainly due to osmotic effects which reduce water availability for plants [14], toxic effects caused by $Cl^-$ and $Na^+$ ions [12], and the nutrient imbalance that imposes deficiency of essential nutrients ($Ca^{2+}$, $Mg^{2+}$, $K^+$, and $NO_3^-$) due to the ion competition caused by the excess of chloride and sodium [15].

Given this limiting scenario, some alternatives have been studied, aiming to mitigate the effects caused by salts on plants, including fertilization management, which seeks compensatory mechanisms for the nutrient deficiency caused by the interaction of toxic ions ($Cl^-$ and $Na^+$) with essential nutrients and, consequently, their accumulation in plant tissues [11,12,16]. From this perspective, Sá et al. [4] obtained improvements in the growth, physiology, and production of the West Indian cherry cv. 'BRS Jaburu' during the first crop cycle, when the recommended nitrogen and phosphorus levels were increased by 40% in plants irrigated with water with an electrical conductivity (ECw) of 3.0 dS m$^{-1}$. Similarly, Lacerda et al. [9] observed that a combination of 70 and 50% of recommended concentrations of nitrogen and potassium, respectively, reduced the effects of salt stress on the anthocyanin and ascorbic acid contents of West Indian cherry fruits. Therefore, irrigation with saline water and NPK fertilization combinations can mitigate the deleterious effects of salt stress on the production and post-harvest quality of West Indian cherry.

From this perspective, this study aimed to evaluate the effect of differing combinations of nitrogen, phosphorus, and potassium on the production and post-harvest quality of West Indian cherry grown under salt stress in the second year of production.

## 2. Materials and Methods

The experiment was conducted using drainage lysimeters under greenhouse conditions at the Agricultural Engineering Academic Unit of the Federal University of Campina Grande in Campina Grande, Paraíba, Brazil (coordinates: 7°15′18″ S, 35°52′28″ W, elevation of 550 m a.s.l.). Meteorological data observed during the experimental period are shown in Figure 1.

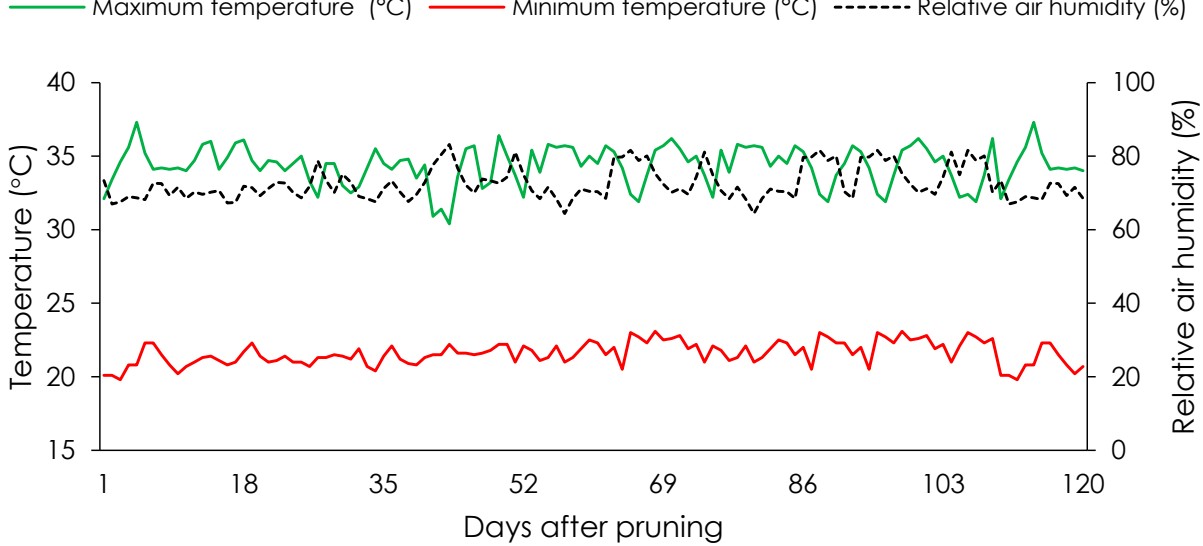

**Figure 1.** Meteorological data observed during the experimental period.

The treatments were distributed in a $2 \times 10$ factorial arrangement following a randomized block design referring to two levels of electrical conductivity in irrigation water (ECw), 0.6 and 4.0 dS m$^{-1}$, and 10 combinations of fertilization (C) with nitrogen, phosphorus, and potassium (NPK) with three replications, totaling 60 experimental units. The combinations studied were: C$_1$ = 80-100-100%; C$_2$ (control) = 100-100-100%; C$_3$ = 120-100-100%; C$_4$ = 140-100-100%; C$_5$ = 100-80-100%; C$_6$ = 100-120-100%; C$_7$ = 100-140-100%; C$_8$ = 100-100-80%; C$_9$ = 100-100-120%; and C$_{10}$ = 100-100-140%. These combinations were based on the fertilization recommendation of N-P$_2$O$_5$-K$_2$O suggested by Cavalcante [17] for the second crop cycle.

The control fertilization combination (100-100-100%) corresponded to the annual application of 100, 60, and 60 g of N, P$_2$O$_5$, and K$_2$O, respectively, per plant as per recommendation. On the other hand, the saline treatments used in the present assay were based on the study by Silva et al. [18]. In March 2020, grafted West Indian cherry seedlings (using rootstocks and scions of the cultivars Junco and Flor Branca, respectively) were acquired from a commercial plant nursery registered with the National Registry of Seeds and Seedlings, located in the São Gonçalo District, Sousa, Paraíba, Brazil. At the end of the first experimental year, the West Indian cherry plants were subjected to water stress for 15 days, after which the pruning of the first cycle was performed and the second production year began. The electrical conductivity levels of the irrigation water and NPK fertilization combinations used in the second production year were the same as in the first year.

The seedlings were transplanted to 200-L drainage lysimeters whose bottoms were covered with geotextile fabric and filled with a layer of 1.0 kg of gravel and 239 kg of Entisol collected from the 0–20 cm soil layer in the municipality of Riachão do Bacamarte—PB (7°15′34″ S and 35°40′1″ W). The physicochemical attributes of the soil were determined according to Teixeira et al. [19], and the results are shown in Table 1. The soil possesses a sandy clay loam texture, with adequate levels of P, K, Ca, and Mg and satisfactory pH and organic matter content. At planting, the soil was supplied with 20 g P$_2$O$_5$ and 20 g K$_2$O as a basal dose.

**Table 1.** Chemical and physical attributes of the soil (0–20 cm depth) used in the experiment.

| Chemical Attributes | | | | | | | | |
|---|---|---|---|---|---|---|---|---|
| **pH H$_2$O** 1:2.5 | **O.M.** g dm$^{-3}$ | **P** mg dm$^{-3}$ | **K$^+$** | **Na$^+$** | **Ca$^{2+}$** cmol$_c$ kg$^{-1}$ | **Mg$^{2+}$** | **Al$^{3+}$ + H$^+$** | |
| 6.5 | 8.1 | 79 | 0.24 | 0.51 | 14.90 | 5.40 | 0.90 | |

| Chemical Attributes | | | | | | Physical Attributes | | | | |
|---|---|---|---|---|---|---|---|---|---|---|
| **EC$_{se}$** | **CEC** | **SAR$_{se}$** | **ESP** | **SB** | **V** | **Particle Fraction** g kg$^{-1}$ | | | **Moisture Content** dag kg$^{-1}$ | |
| dS m$^{-1}$ | cmol$_c$ kg$^{-1}$ | (mmol L$^{-1}$)$^{0.5}$ | % | cmol$_c$ kg$^{-1}$ | % | Sand | Silt | Clay | 33.42 kPa [1] | 1519.5 kPa [2] |
| 2.15 | 21.95 | 0.16 | 2.3 | 21.05 | 95.89 | 572.7 | 100.7 | 326.6 | 25.91 | 12.96 |

pH—potential of hydrogen; O.M.—organic matter: Walkley–Black Wet digestion; Ca$^{2+}$ and Mg$^{2+}$—extracted with 1 M KCl at pH 7.0; Na$^+$ and K$^+$—extracted with NH$_4$OAC 1 M at pH 7.0; Al$^{3+}$ + H$^+$—extracted with CaOAc 0.5 M at pH 7.0; EC$_{se}$—electrical conductivity of the saturation extract; CEC—cation exchange capacity; SAR$_{se}$—sodium adsorption ratio of the saturation extract; ESP—exchangeable sodium percentage; SB—sum of bases (K$^+$ + Ca$^{2+}$ + Mg$^{2+}$ + Na$^+$); V—base saturation ([SB/CEC] $\times$ 100); [1–2]—refers to field capacity and the permanent wilting point, respectively.

With regard to the saline treatments, water with the electrical conductivity levels of 0.6 and 4.0 dS m$^{-1}$ were prepared by adding the salts NaCl, CaCl$_2$.2H$_2$O, and MgCl$_2$.6H$_2$O in a respective proportion of 7:2:1 to water available for irrigation in the study region (ECw = 0.38 dS m$^{-1}$), using the relationship between the ECw and the concentration of salts proposed by Richards [20], according to Equation (1). The electrical conductivity of water was monitored and adjusted periodically before irrigation.

$$Q \approx 640 \times ECw \tag{1}$$

where Q = sum of cations (mg L$^{-1}$) and ECw = water electrical conductivity (dS m$^{-1}$).

The treatments with saline water began to be applied 30 days after the seedlings were transplanted into the lysimeters, following a two-day irrigation schedule. Each lysimeter received water according to the respective treatment, aiming to maintain soil moisture close to field capacity in all experimental units, which was estimated according to the soil–water balance determined by Equation (2).

$$VI = \frac{(Va - Vd)}{(1 - LF)} \qquad (2)$$

where VI = water to be applied in the irrigation event (mL), Va = water volume applied in the previous irrigation event (mL), Vd = water volume drained after the previous irrigation event (mL), and LF = 0.10 leaching fraction, applied every 90 days to prevent excessive salt accumulation in the soil solution.

The applications with the fertilization combinations containing N, P, and K were split into 24 applications, always applied via topdressing, at 15-day intervals. The NPK sources used were calcium nitrate, monoammonium phosphate, and potassium sulfate, respectively. The fertilization interventions with micronutrients were performed every 15 days via foliar application on the adaxial and abaxial leaf surfaces by applying a solution containing 1.0 g L$^{-1}$ of Dripsol® (Mg = 1.1%; Zn = 4.2%; B = 0.85%; Fe = 3.4%; Mn = 3.2%; Cu = 0.5%; and Mo = 0.05%). With regard to the crop management practices, first-cycle pruning, manual hoeing, soil scarification, and phytosanitary control were performed whenever necessary during the experimental period. For the second year of West Indian cherry production, the NPK combinations were adapted following the recommendations of Cavalcante [17], consisting of 200, 30, and 80 g of N, P$_2$O$_5$, and K$_2$O, respectively, per year, per experimental unit. Therefore, in the second year of production, the application of treatments began 15 days after first-cycle pruning (DAP) by maintaining the irrigation and phytosanitary managements used in the first year.

Harvest occurred when the fruits showed a red color but were still firm enough to withstand handling, corresponding to the commercial maturity stage. The total fruit weight (TFW, g) was quantified as the mass of the fruits harvested daily in each experimental unit during the production period. The total number of fruits (TNF) was obtained by counting all fruits harvested per plant in each plot. The mean fruit weight (MFW, g) was obtained by dividing the total fruit weight by the total number of fruits harvested per plant. Furthermore, the polar (PFD—mm) and equatorial diameter (EFD—mm) of the fruits from each plot were evaluated individually by randomly choosing 20 fruits per plant. This procedure was performed using a digital caliper and measuring the distance from the apex to the base of the fruit (PFD) and in the equatorial region (EFD).

After analyzing the production variables, the fruits were sorted to eliminate those with damage or impurities. Then the fruits were immersed for 15 min in chlorinated water (50 ppm) and washed in running water. The pulp was obtained by grinding the fruits in an industrial blender (BecKer®, modelo RBT-6) and passing them through a sieve to remove the uncrushed fiber portions and other residues. Then the pulp was stored in polyethylene bags and stored in a freezer at a controlled temperature of −18 °C until the analyses.

*Characterization of the Fruit Pulp*

The physicochemical parameters of the West Indian cherry pulp were analyzed in triplicate, according to the methodology of the Adolfo Lutz Institute [21], by determining the total soluble solids (TSS, °Brix) using a portable refractometer (Euro Analytical, RZT model); the potential of hydrogen (pH) through direct reading in the samples, using a digital potentiometer (model MB11, MS Techonopon®, Piracicaba, SP, Brazil) previously calibrated with buffer solutions at pH 4.0 and 7.0; the titratable acidity (TA, % citric acid), obtained by titration with 0.1 mol L$^{-1}$ NaOH until reaching a pH of 8.2–8.4. The maturity ratio (RAT = TSS/TA) was obtained from the quotient between the total soluble solids and

the titratable acidity, whereas the vitamin C content (VTC, mg ascorbic acid $100 \text{ g}^{-1}$) was estimated according to Oliveira et al. [22].

The reducing sugars (RSU, g $100 \text{ g}^{-1}$) were determined using dinitrosalicylic acid [23] (Miller, 1959). The total phenolic compounds (TPC, mg $100 \text{ g}^{-1}$) were quantified with Folin–Ciocalteu reagent [24]. The total anthocyanins (ANT, mg $100 \text{ g}^{-1}$) and flavonoids (FLA, mg $100 \text{ g}^{-1}$) were analyzed according to the recommendations of Francis [25]. All readings were performed in a spectrophotometer (BEL Photonics, model SP1102) according to their respective wavelengths.

The data obtained were tested for normality of distribution (Shapiro–Wilk test). Then analysis of variance was performed using the F-test ($p \leq 0.05$) for the electrical conductivity levels of the irrigation water. In turn, the fertilization combinations (NPK) were compared using the Scott–Knott clustering test. All statistical analyses were performed using the software program Sisvar [26].

## 3. Results and Discussion

According to the summary of the analyses of variance (Table 2), there was a significant interaction ($p \leq 0.01$) between the ECw and the NPK fertilization combinations for the total number of fruits (TNF), total fruit weight (TFW), and the mean fruit weight (MFW). On the other hand, the ECw had a significant isolated effect ($p \leq 0.01$) on the equatorial fruit diameter (EFD) 140 days after pruning at the end of first cycle, during the second year of production.

**Table 2.** Summary of the analysis of variance for the total number of fruits (TNF), total fruit weight (TFW), mean fruit weight (MFW), polar diameter (PFD), and equatorial diameter (EFD) of West Indian cherry irrigated with water having different electrical conductivity levels and fertilized with combinations of nitrogen, phosphorus, and potassium during the second year of production, 140 days after pruning (DAP).

| SV | DF | Mean Squares | | | | |
| --- | --- | --- | --- | --- | --- | --- |
| | | TNF | TFW | MFW | PFD | EFD |
| Water electrical conductivity—ECw | 1 | 94318.627 ** | 5667555.518 ** | 15.347995 ** | 1.532482 ns | 18.734329 ** |
| Fertilization combinations—C | 9 | 38502.571 ** | 819854.9348 ** | 1.283977 ** | 2.652434 ns | 4.346666 ns |
| Interaction (ECw × C) | 9 | 35052.094 ** | 1037320.541 ** | 0.739933 * | 5.853079 ns | 5.084948 ns |
| Block | 2 | 2179.8871 ns | 78568.0114 ns | 0.110357 ns | 10.996305 ns | 6.016640 ns |
| Residual | 38 | 1286.6893 | 73725.2114 | 0.329977 | 3.553961 | 2.675142 |
| CV (%) | | 12.91 | 21.89 | 12.98 | 11.08 | 8.66 |

SV—Source of variation; DF—degree of freedom; CV (%)—coefficient of variation; ns, *, **, not significant and significant at $p \leq 0.05$ and $p \leq 0.01$, respectively.

According to the expansion of the data for the TNF of West Indian cherry (Figure 2), the levels of ECw (0.6 and 4.0 dS $\text{m}^{-1}$) did not influence this variable significantly when the plants were subjected to fertilization combinations $C_2$ (100-100-100%), $C_6$ (100-120-100%), $C_7$ (100-140-100%), and $C_8$ (100-100-80%).

The highest values of fruit production were observed in plants irrigated with low-salinity water (0,6 dS $\text{m}^{-1}$) and treated with fertilization combinations $C_5$ (602.2), $C_1$ (473.5), $C_4$ (414.9), and $C_3$ (383.8), differing statistically from each other ($p \leq 0.01$) and from the plants irrigated with the ECw of 4.0 dS $\text{m}^{-1}$ and treated with combinations $C_5$ (269), $C_1$ (284.1) $C_4$ (133.9), and $C_3$ (243.1), respectively. On the other hand, the plants irrigated with the ECw of 4.0 dS $\text{m}^{-1}$ showed the highest fruit production values when treated with combinations $C_9$, $C_8$, $C_5$, and $C_1$, but only the plants which received the $C_9$ combination differed ($p \leq 0.01$) from the plants irrigated with the ECw 0.6 dS $\text{m}^{-1}$.

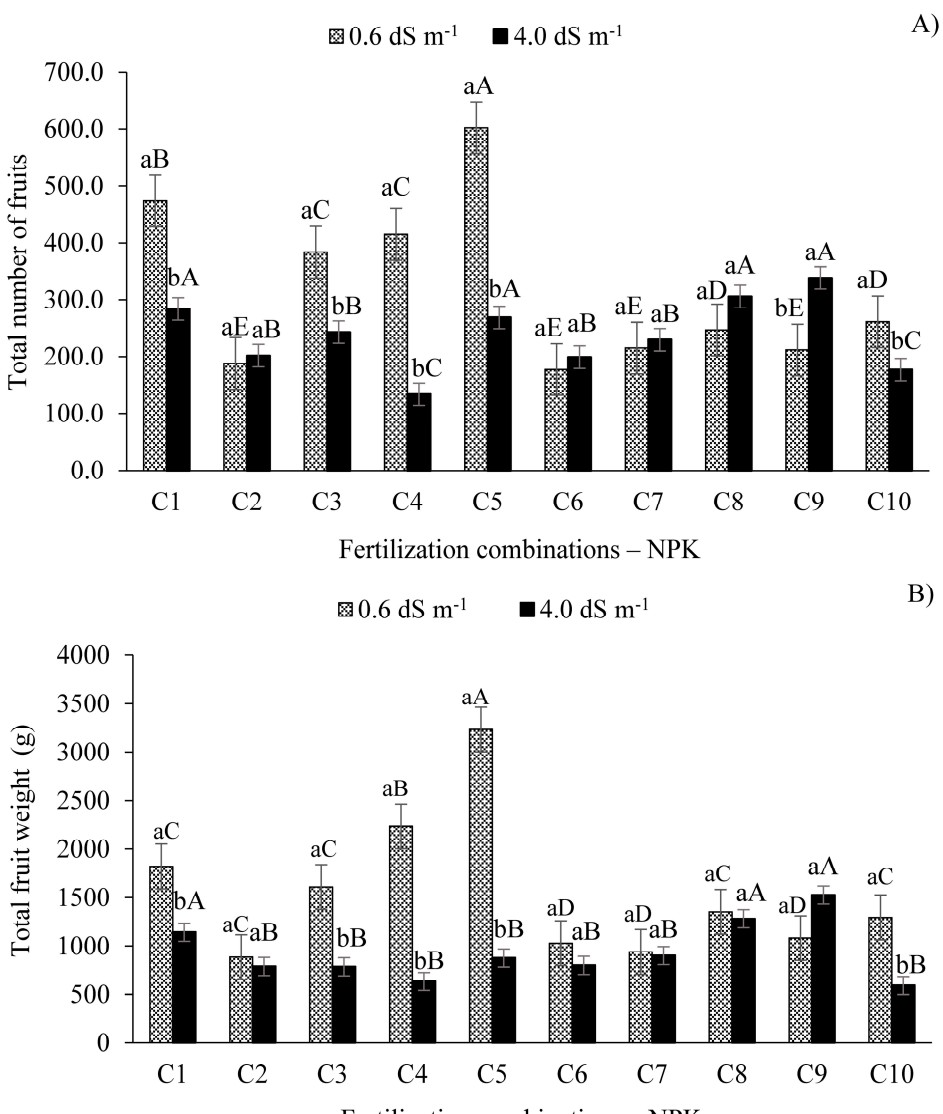

**Figure 2.** Expansion of the interaction between levels of ECw and NPK fertilization combinations for the total number of fruits (**A**) and total fruit weight (**B**) of West Indian cherry 140 days after pruning during the second year of production. $C_1$ = 80-100-100; $C_2$ = 100-100-100; $C_3$ = 120-100-100; $C_4$ = 140-100-100; $C_5$ = 100-80-100; $C_6$ = 100-120-100; $C_7$ = 100-140-100, $C_8$ = 100-100-80, $C_9$ = 100-100-120, and $C_{10}$ = 100-100-140% of the recommended N-$P_2O_5$-$K_2O$ level; means followed by identical uppercase letters indicate that, for the same type of water, there are no significant differences between the fertilization combinations according to the Scott–Knott test at $p > 0.05$ of probability, whereas identical lowercase letters in the same fertilization combination indicate no significant difference between salinity levels (F-test, $p > 0.05$). Means of three replicates ± standard error.

The plants irrigated with the ECw of 0.6 dS m$^{-1}$ and fertilized with the combination containing 20% less phosphorus than the recommendation ($C_5$ = 100-80-100) produced the highest TNF compared to the other combinations, with mean increases of 68.75% and 70.55% in relation to the plants fertilized with the recommended NPK level ($C_2$ = 100-100-100%) and those that received 20% more phosphorus ($C_6$ = 100-120-100%), which produced the lowest TNF.

The plants irrigated with the ECw of 4.0 dS m$^{-1}$ and fertilized with the combination with a 20% increase in potassium ($C_9$ = 100-100-120%) produced the highest TNF (338.5) but did not differ statistically from those fertilized with combinations $C_1$, $C_5$, and $C_8$. However, combination $C_9$ produced the highest TNF ($p \leq 0.01$) when paired with the lower electrical conductivity water. For combinations $C_2$, $C_6$, $C_7$, and $C_8$, despite the absence of a significant

difference in effect between each other, there were higher fruit production values in relation to the water with the lower electrical conductivity. On the other hand, the lowest number of fruits (133.9) in this study was observed when the plants received water with the electrical conductivity level of 4.0 dS m$^{-1}$ and fertilization combination $C_4$ (140-100-100%), although they do not differ from combination $C_{10}$ (177.3) (Figure 2A).

Furthermore, the increase in the ECw to 4.0 dS m$^{-1}$ reduced the TNF by 25%. However, in the plants under salt stress treated with combination $C_9$ (100-100-120%), there was an increase of 40.38% in the TNF when paired with the recommended fertilization level ($C_2$) and of 60.45% compared to the combination with a 40% N increase ($C_4$ = 140-100-100), which showed the lowest TNF. The TNF reduction in plants under salt stress is possibly associated with salt accumulation in the soil, leading to a reduction in the osmotic potential of the soil solution and increasing the energy expenditure of the plant to absorb water and nutrients [27]. Therefore, these effects limit physiological and biochemical processes in the plant, restricting cell division and elongation and, consequently, reducing growth and fruit production [28], as observed in this study.

A similar behavior was observed for the total fruit weight (TFW), with the ECw levels studied not significantly affecting this variable when treated with fertilization combinations $C_2$, $C_6$, $C_7$, $C_8$, and $C_9$ (Figure 2B). For the ECw of 0.6 dS m$^{-1}$, the highest TFW values were found when plants were treated with combinations $C_5$, $C_4$, $C_1$, $C_3$, and $C_{10}$, thus differing ($p \leq 0.01$) from the plants that received ECw of 4.0 dS m$^{-1}$. Furthermore, the pairing of the ECw of 0.6 dS m$^{-1}$ and the recommended N-P-K fertilization combination ($C_2$) generated the lowest TFW (885.16 g), on average 72.63% lower than the TFW observed in plants treated with combination $C_5$.

The higher electrical conductivity irrigation water decreased the TFW by 39.71%. These results agree with those observed by Sá et al. [29], who observed a reduction of 217.48 g (20.3%) per plant in sugar apple (*Anonna squamosa* L.) plants subjected to ECw of 3.0 dS m$^{-1}$ compared to those irrigated with 0.8 dS m$^{-1}$. For West Indian cherry, Silva et al. [28] observed that the increase in the ECw from 0.3 to 4.3 dS m$^{-1}$ promoted a linear reduction of 3.95% in the mean fruit weight per unit. On the other hand, the plants irrigated with water having an electrical conductivity of 4.0 dS m$^{-1}$ and treated with combination $C_9$ had TFW increases of 48.20% and 61.15%, compared to combinations $C_2$ and $C_{10}$ (which had lowest TFW values), respectively (Figure 2B).

The ECw of 4.0 dS m$^{-1}$, when paired with combination $C_9$, resulted in the highest total fruit weight (1525.3 g), but did not yield any difference from the value obtained with the lower electrical conductivity water paired with this fertilization combination. However, the lowest total fruit weight (592.57 g) was observed when the plants received the treatment with the higher ECw and fertilization combination $C_{10}$, not differing from the other treatments, except for $C_1$, $C_8$, and $C_9$ (Figure 3).

The results of this study indicate that, in the cultivation of West Indian cherry with lower electrical conductivity irrigation water (0.6 dS m$^{-1}$), the combination containing 20% less of the recommended phosphorus level provided greater nutritional balance, favoring physiological and biochemical processes in plants [27,30], which directly influenced the number and total weight of West Indian cherry fruits (Figure 2A,B). The importance of these results should be highlighted as they mean a reduction in the production cost of this crop in semi-arid regions.

At adequate levels, phosphorus favors root development; improves water-use efficiency, nutrient uptake, and utilization by plants; and assists in plant respiration, photosynthesis, and energy release for metabolic reactions [12,31]. Accordingly, Bezerra et al. [10] observed a reduction of 28.13% in the number of fruits in the second production cycle of guava due to the increase in the electrical conductivity of irrigation water from 0.3 to 3.5 dS m$^{-1}$ combined with nitrogen fertilization. Ferreira et al. [12] also observed a 20.80% reduction in the number of sugar apple fruits with the increase in water salinity from 0.8 to 3.0 dS m$^{-1}$ and fertilization with NPK.

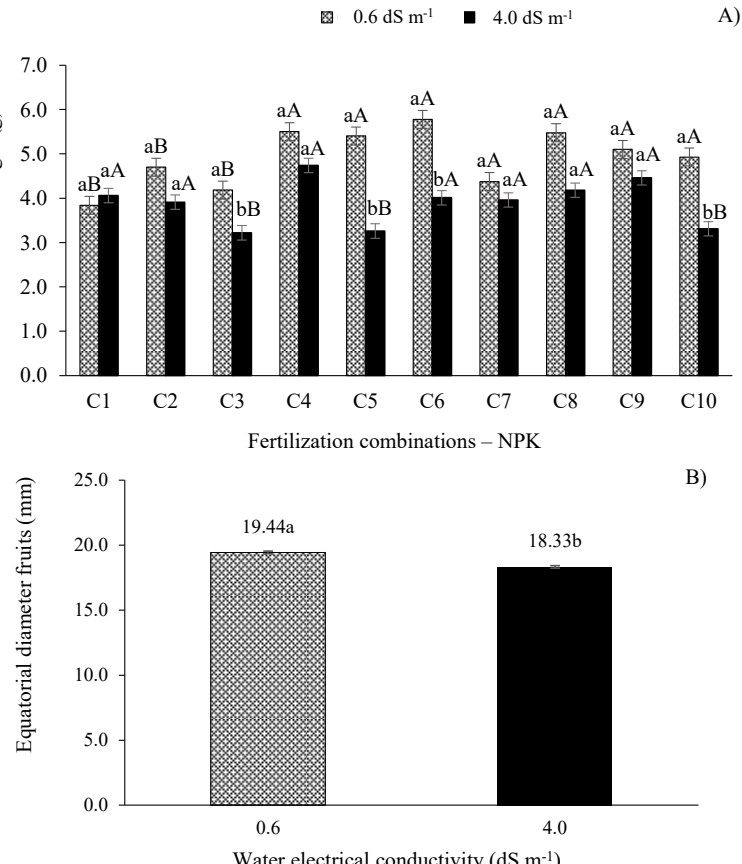

**Figure 3.** Mean fruit weight (**A**) and equatorial fruit diameter (**B**) of West Indian cherry as a function of the interaction between ECw and NPK fertilization combinations 140 days after pruning during the second year of production. For description of fertilizer combinations (C), see Figure 2. Means followed by identical uppercase letters indicate that, for the same type of water, there are no significant differences between fertilization combinations found by the Scott–Knott test at $p > 0.05$ probability. Identical lowercase letters in the same fertilization combination indicate no significant difference between salinity levels (F-test, $p > 0.05$). Mean of three replicates ± standard error.

On the other hand, the TNF increase associated with combinations $C_6$, $C_7$, $C_8$, and $C_9$ and the TFW increase associated with combination $C_9$ in plants subjected to salt stress observed in the present study are probably related to the increase in K availability (20%), since this nutrient acts in osmotic regulation and in the activation of several enzymes that act in plant respiration and photosynthesis, directly affecting fruit production [29]. Furthermore, this effect could be related to the physiological role played by potassium in osmoregulation, intimately associated with the regulation of the cell osmotic potential and, consequently, contributing to homeostasis and fruit production [32]. From this perspective, Lima et al. [33] observed the mitigation of the negative effects of salinity on the total number of fruits and fresh fruit mass of West Indian cherry through potassium fertilization.

With regard to the mean fruit weight (Figure 3A), the plants irrigated with the ECw of 0.6 dS m$^{-1}$ showed significant differences ($p \leq 0.01$) compared to those irrigated with the ECw of 4.0 dS m$^{-1}$ when paired with combinations $C_3$, $C_5$, $C_6$, and $C_{10}$. Furthermore, the plants irrigated with the ECw of 0.6 dS m$^{-1}$ and treated with combinations $C_4$, $C_5$, $C_6$, $C_7$, $C_8$, $C_9$, and $C_{10}$ obtained the highest MFW without differing statistically among each other, but did differ ($p \leq 0.01$) from plants treated with combinations $C_1$, $C_2$, and $C_3$.

Similar to TFW, the treatment with the salinity level of 0.6 dS m$^{-1}$ and fertilization combination $C_5$ increased the MFW by 13.01% and 28.96% compared to the plants that received the recommended N-P-K combination ($C_2$ = 100-100-100%), and combination $C_1$ (80-100-100%) which showed the lowest MFW value (3.84 g). When applied at proper

levels, phosphorus plays a key role in plant metabolism, especially due to its role in energy storage, which is vital for metabolic and structural functions such as photosynthesis [29]. Therefore, this result can be attributed to the reduction by 20% in the supplementation of $P_2O_5$ and the adequate N and $K_2O$ levels present in combination $C_6$ (100-80-100%). Thus, even with West Indian cherry being a local regional plant, it easily adapts to various types of soil, highlighting the importance of properly managing fertilization and plant nutrition, especially with macronutrients such as P [33].

However, despite the MFW reductions in plants irrigated with the higher electrical conductivity (4.0 dS m$^{-1}$) water, the 40% increase in the recommended nitrogen level ($C_4$ = 140-100-100%) increased the MFW by 17.51% compared to the recommended N-P-K levels ($C_2$), differing statistically ($p \leq 0.01$) from plants treated with combinations $C_3$, $C_5$, and $C_{10}$.

Among the strategies used to mitigate the effects caused by salts, the management of fertilization has promoted positive effects on plant responses, especially when the fertilizer in focus is nitrogen-based. This effect can be justified by the role played by nitrogen in plants, as it is present in several biomolecules, including amino acids, which act in promoting osmotic adjustment and in the mitigation of the effects of reactive oxygen species [34].

Nitrogen is extremely important to plants as it acts in various structural functions and is part of vital organic compounds, e.g., amino acids, chlorophylls, proline, nucleic acids, proteins, etc. [13,27]. These functions are essential since they contribute to increasing the osmotic adjustment capacity of plants under water and salt stress [9,34]. From this perspective, Sá et al. [35] observed that N levels 40% above recommendation for West Indian cherry were sufficient to mitigate the harmful effects caused by the ECw of 3.0 dS m$^{-1}$.

As observed in Figure 3B, the equatorial fruit diameter (EFD) of West Indian cherry decreased significantly ($p \leq 0.01$) as a function of the increase in the ECw, regardless of the fertilization combination. Plants irrigated with the higher ECw (4.0 dS m$^{-1}$) showed EFD reductions of 5.71% (1.11 mm) compared to those subjected to the lower ECw (0.6 dS m$^{-1}$).

Therefore, the EFD reduction observed in plants irrigated with the ECw of 4.0 dS m$^{-1}$ could be related to the damage caused by the excess of salts in the cell membrane and induced by reactive oxygen species [27]. Overall, salt stress causes different types of damage to plants due to the reduction in the osmotic potential of the soil solution, the nutrient imbalance caused by the excess of salts, and the disturbance in plant metabolism caused by the high concentration of toxic ions in the protoplasm, especially Cl$^-$ and Na$^+$ ions [36].

Therefore, the set of these factors limits the uptake of essential nutrients, affecting plant growth, development, and production [9]. Therefore, it is essential to adopt tools with the potential to mitigate the effects of water salinity and use plant species with higher salt tolerance [12]. An excess of salts in soil solution causes various types of damage to plants, including changes in stomatal conductance [27], changes in the quantum efficiency of photosystem II [8], and nutrient imbalance, which affects plant development [32], causes the de-structuration of membrane permeability [7], and restricts photosynthesis [16].

According to the summary of the analyses of variance shown in Table 3, there was a significant interaction ($p \leq 0.01$) between ECw levels and N-P-K fertilization combinations (ECw × C) on all post-harvest variables of West Indian cherry studied during the second production cycle.

The results shown in Figure 4A reveal that the titratable acidity (% citric acid) of West Indian cherry plants irrigated with ECw of 4.0 dS m$^{-1}$ differed ($p \leq 0.01$) from those grown with ECw of 0.6 dS m$^{-1}$, except in plants that received fertilization combinations $C_{10}$ and $C_5$. Furthermore, the TA increased ($p \leq 0.01$) in plants irrigated with the higher salinity water (4.0 dS m$^{-1}$) paired with fertilization combinations $C_1$, $C_3$, $C_4$, $C_6$, $C_7$, and $C_8$.

**Table 3.** Summary of the F-test for total titratable acidity, potential of hydrogen, total soluble solids, TSS/TA ratio, vitamin C, reducing sugars, total phenolic compounds, flavonoids, and anthocyanins in West Indian cherry irrigated with saline water and treated with different nitrogen, phosphorus, and potassium combinations (C) during the second year of production 140 days after the first-cycle pruning.

| SV | F-Test | | | | | | | | |
| --- | --- | --- | --- | --- | --- | --- | --- | --- | --- |
| | **TA** | **pH** | **TSS** | **RAT** | **VTC** | **RSU** | **TPC** | **FLA** | **ANT** |
| Water electrical conductivity—ECw | ** | ** | ** | ** | ** | ** | ns | ** | ns |
| Fertilization combinations—C | ** | ** | ** | ** | ** | ** | ** | ** | ** |
| Interaction (ECw × C) | ** | ** | ** | ** | ** | ** | ** | ** | ** |
| Block | ns | ns | ns | ns | ns | ns | ns | ns | ns |
| CV (%) | 0.71 | 0.22 | 1.32 | 1.49 | 2.46 | 3.70 | 3.01 | 3.63 | 3.56 |
| General mean | 1.78 | 3.36 | 9.63 | 5.39 | 3746 | 3030 | 2953 | 6.94 | 6.06 |

SV—Source of variation; CV (%)—coefficient of variation; ns, **, indicate not significant and significant by the F-test at ≤ 0.01, respectively.

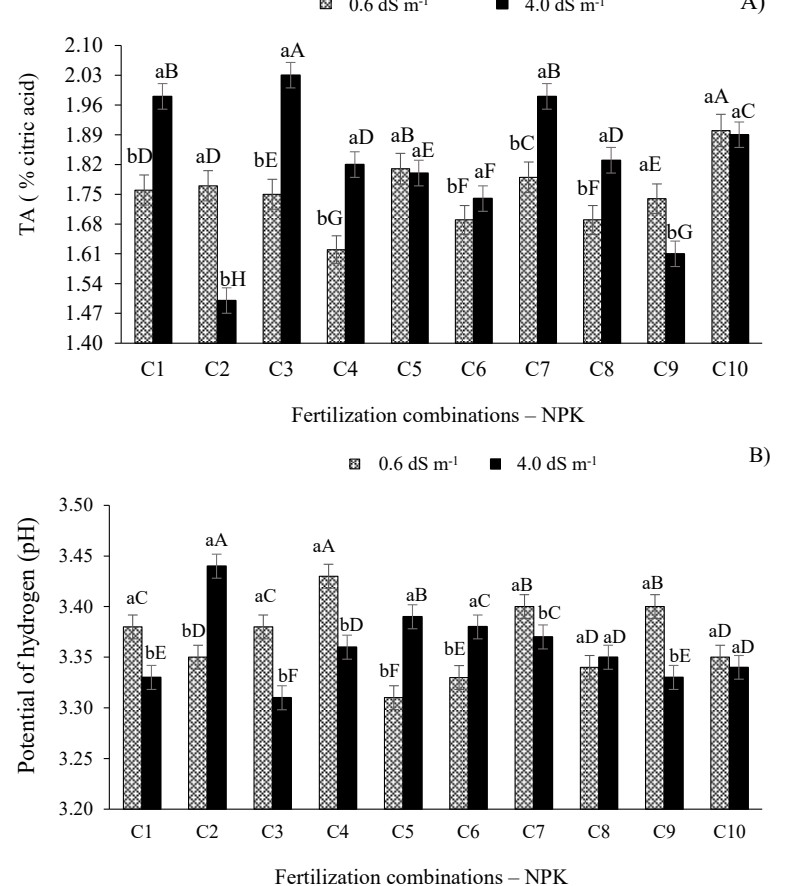

**Figure 4.** Titratable acidity (**A**) and potential of hydrogen (**B**) of West Indian cherry pulp as a function of the interaction between the ECw and combinations of NPK fertilization in the second year of production. For description of fertilizer combinations (C), see Figure 2. Means followed by identical uppercase letters indicate that, for the same type of water, there are no significant differences between fertilization combinations by the Scott–Knott test at $p > 0.05$ probability. Identical lowercase letters in the same fertilization combination indicate no significant difference between salinity levels (F-test, $p > 0.05$). Mean of three replicates ± standard error.

The saline treatment with the ECw of 4.0 dS m$^{-1}$ paired with combination C$_3$ (120-100-100%) produced the highest titratable acidity in the fruit pulp, differing ($p \leq 0.01$) from other combinations. The TA increased by 12.81% and 26.11% compared to the plants of the control treatment (C$_2$—100-100-100% of the recommended N-P-K level) when the plants were irrigated with the ECw of 0.6 and 4.0 dS m$^{-1}$, respectively.

The results of the titratable acidity of West Indian cherry pulp (Figure 4A) highlight its suitability for the processing of concentrated juice [16] and are important because the consumption of this fruit usually occurs through pulp, with the increase in the citric acid levels increasing the quality of the final product as this compound acts as an antioxidant and reduces the need for pulp acidifiers [10]. Adriano et al. [37] observed that, in ripe and semi-ripe West Indian cherry fruits, the TA was 3.15% and 3.26%, respectively.

In a study of the West Indian cherry cultivar 'BRS 366 Jaburu' irrigated with water having electrical conductivity levels ranging from 0.6 to 3.8 dS m$^{-1}$ and treated with phosphate fertilization, Lima et al. [7] obtained fruits with TA values ranging from 1.20% to 1.80%, i.e., values close to the mean values obtained in the present study, 1.75% and 1.82% citric acid for the ECw of 0.6 and 4.0 dS m$^{-1}$, respectively. Furthermore, these TA values are significantly higher than the 0.94% obtained by Moura et al. [38] when studying the post-harvest quality of fruits of the West Indian cherry cv. Flor Branca and the 0.80% recommended by Brazilian regulations [39].

For the pulp pH of West Indian cherry, the treatments with different electrical conductivity-level water differed ($p \leq 0.01$) for all fertilization combinations except C$_8$ and C$_{10}$ (Figure 4B). There was a significant increase in pulp pH in treatments irrigated with the ECw of 4.0 dS m$^{-1}$ and paired with fertilization combinations C$_2$ (control), C$_5$, and C$_6$ in relation to the other combinations at the same salinity level (Figure 4B).

Given the results shown in Figure 4B, it can be seen that the pulp pH of West Indian cherry decreased as a function of the increase in irrigation water salinity, thereby increasing fruit acidity. The pulp pH value is extremely important as it represents the degree of pulp deterioration. According to Brazilian regulations [39], pulp pH values lower than 4.5 are ideal to prevent the proliferation of microorganisms and maintain pulp quality. Adriano et al. [37] analyzed the fruit quality of the West Indian cherry cv. Olivier and obtained pH values of 3.68 in ripe fruits and 3.56 in semi-ripe fruits, i.e., close to the values of the present study.

Lima et al. [7] studied the effects of ECw levels ranging from 0.6 to 3.8 dS m$^{-1}$ and phosphate fertilization on the physicochemical qualities of West Indian cherry and also observed a linear reduction in the pulp pH, with a reduction of 3.02% per unit increase in the electrical conductivity of irrigation water. On the other hand, Silva et al. [16] studied the fruit quality of the West Indian cherry cv. Flor Branca irrigated with saline water (ECw ranging from 0.30 to 4.30 dS m$^{-1}$) and fertilized with combinations of nitrogen and potassium and did not observe significant effects on the pulp pH.

According to the results shown in Figure 5A, the plants irrigated with the electrical conductivity of 4.0 dS m$^{-1}$ showed significant differences ($p \leq 0.01$) in the total soluble solids (°Brix) in the pulp in relation to the plants grown with the ECw of 0.6 dS m$^{-1}$, except for those treated with combinations C$_2$ and C$_3$. The treatments with combinations C$_1$ and C$_{10}$ paired with the ECw of 4.0 dS m$^{-1}$ produced the highest TSS values, differing ($p \leq 0.01$) from the other fertilization combinations. The highest values of total soluble solids in West Indian cherry pulp are related to treatment with ECw of 4.0 dS m$^{-1}$ and combinations C$_1$ and C$_{10}$, which were, on average, 21.10% higher than pulp from plants which received treatment with the recommended fertilization combination (C$_2$) and the same salinity treatment (Figure 5A).

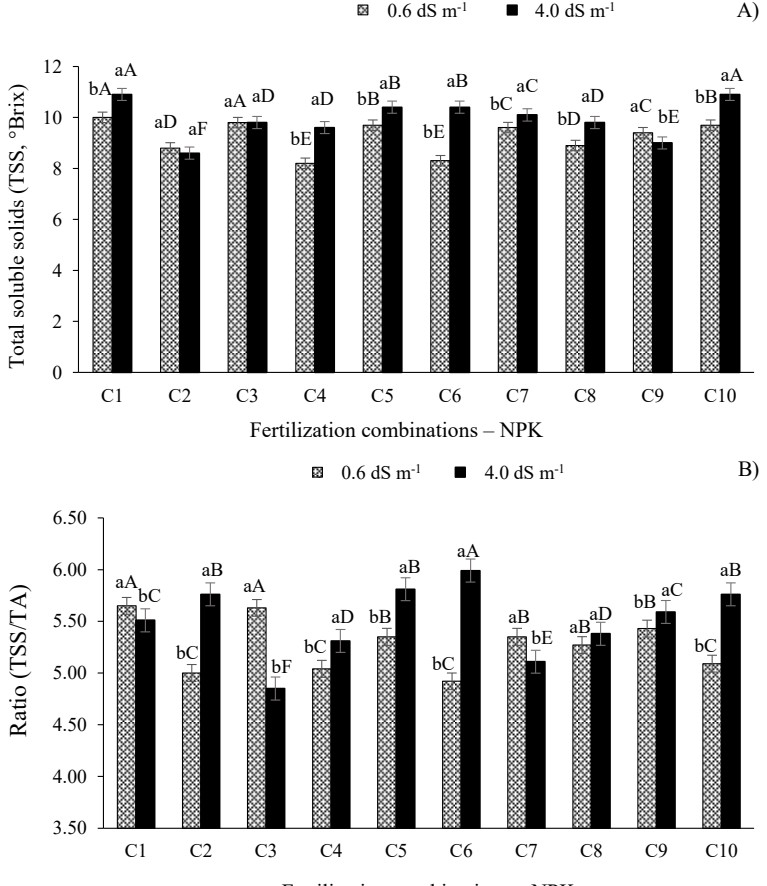

**Figure 5.** Total soluble solids (**A**) and maturity index ratio—TSS/TA (**B**) of West Indian cherry pulp as a function of the interaction between the ECw and combinations of NPK fertilization evaluated in the second year of production. For description of fertilizer combinations (C) see Figure 2. Means followed by identical uppercase letters indicate that, for the same type of water, there are no significant differences between fertilization combinations by the Scott–Knott test at *p* > 0.05 probability. Identical lowercase letters in the same fertilization combination indicate no significant difference between salinity levels (F-test, *p* > 0.05). Mean of three replicates ± standard error.

Overall, the increase in the ECw levels increased the content of total soluble solids (°Brix) of West Indian cherry pulp (Figure 5A). These results are important for this crop as the fruits are marketed fresh and high °Brix concentrations correspond to high contents of sugars and organic acids [40]. Likewise, Lima et al. [7] observed that the increase in the ECw promoted linear increases in the soluble solids of West Indian cherry, whereas Silva et al. [16] observed no significant differences in the °Brix of West Indian cherry in plants irrigated with different ECw levels. Adriano et al. [37] obtained the respective soluble solid values in ripe and semi-ripe fruits of 7.58 and 7.42 (°Brix). Therefore, the TSS values found in this study are higher than the values of the authors mentioned before.

The significant effects ($p \leq 0.01$) of salt stress were also observed in the TSS/TA ratio (maturity index), except for combination $C_8$ (Figure 5B). Furthermore, the highest TSS/TA ratio value (5.99) was recorded in plants irrigated with the higher electrical conductivity water and treated with combination $C_6$ (100-120-100% of the recommended N-P-K level), differing ($p \leq 0.01$) from the other fertilization combinations. For the plants irrigated with the ECw of 0.6 dS m$^{-1}$, the highest values of this index were observed when paired with combinations $C_1$ (5.65) and $C_3$ (5.63), differing ($p \leq 0.01$) from the other fertilization treatments, while the optimum value is considered 6.8 [41].

The maturity index is represented by the relationship between total soluble solids and titratable acidity, corresponding to one of the main tools used to analyze fruit flavor since

it represents the balance between these two variables [42]. Ferreira et al. [12] evaluated the production and post-harvest quality of sugar apple irrigated with saline water (ECw of 0.8 and 3.0 dS m$^{-1}$) and fertilized with different NPK combinations and observed that the highest values were obtained when using the NPK combination of 125-125-100% of recommended levels. In another study, Silva et al. [16] observed no significant differences in the maturity index of West Indian cherry in plants irrigated with different ECw levels and N-K combinations, while Adriano et al. [37] obtained TSS/TA ratio values of 2.41 and 2.27 from ripe and semi-ripe West Indian cherry fruits, respectively.

According to the expansion of data for the content of vitamin C of West Indian cherry (Figure 6A), there were statistical differences ($p \leq 0.01$) between the electrical conductivity levels studied. It is noted that the higher electrical conductivity of irrigation water (4.0 dS m$^{-1}$) and combination $C_{10}$ promoted the highest value of this variable, which was significantly higher ($p \leq 0.01$) than the other combinations. On the other hand, the plants irrigated with the ECw of 0.6 dS m$^{-1}$ showed highest values of vitamin C when they received combinations $C_2$, $C_6$, $C_7$, and $C_9$, differing only from combinations $C_4$ and $C_8$.

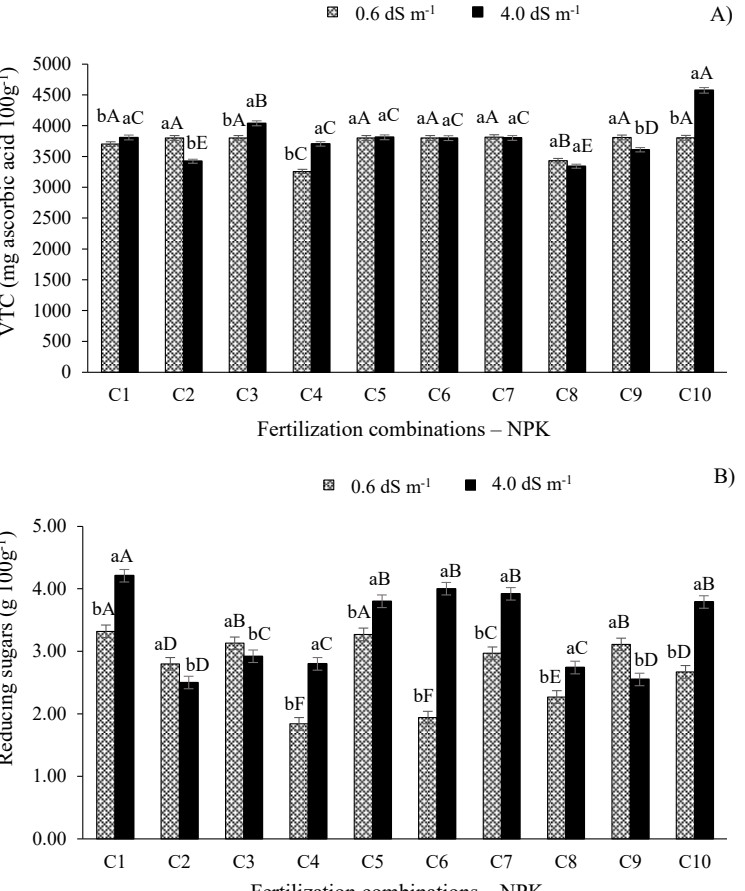

**Figure 6.** Vitamin C (**A**) and reducing sugars (**B**) of West Indian cherry pulp as a function of the interaction between the ECw and NPK fertilization combinations in the second year of production. For description of fertilizer combinations (C) see Figure 2. Means followed by identical uppercase letters indicate that, for the same type of water, there are no significant differences between fertilization combinations by the Scott–Knott test at $p > 0.05$ probability. Identical lower-case letters in the same fertilization combination indicate no significant difference between salinity levels (F-test, $p > 0.05$). Mean of three replicates ± standard error.

Silva et al. [16] studied the production and quality of West Indian cherry as a function of the increase in the ECw and NK combinations and observed a reduction of 2.16% in the vitamin C content of the fruits for every unit increase in the ECw of irrigation water.

This reduction caused by the excess of salts in irrigation water is intimately related to the reduction in the content of sugars with soluble hexoses, responsible for the synthesis of ascorbic acid by fruits [43].

The reducing sugars in the pulp of West India cherry fruits showed significant effects as a function of the ECw (Figure 6B). Despite the increase in the reducing sugars observed in plants irrigated with the ECw of 4.0 dS m$^{-1}$, only those plants fertilized with combination $C_1$ (80-100-100% of the recommended N-P-K level) differed ($p \leq 0.01$) from the other combinations, showing a 68.40% increase compared to the control plants ($C_2$) irrigated with the same salinity. For the treatments with the ECw of 0.6 dS m$^{-1}$, combination $C_1$ also produced the highest result for this variable, not differing ($p > 0.05$) only from combination $C_5$.

Given these results, it can be seen that the supply of adequate N levels in plants under salt stress favors the biosynthetic pathways related to carbohydrates and maintains the balance in the assimilation of N and other essential nutrients, increasing the concentration of organic constituents such as sugars [44]. From this perspective, in both fresh fruits and those used for industrial processing, higher sugar contents are essential as they increase sweetness and viscosity, in addition to improving the texture and decreasing the freezing point of the pulp [45]. Lacerda et al. [9] studied the quality of West Indian cherry fruits in plants irrigated with saline water and fertilized with nitrogen–potassium and observed a linear reduction in the total sugar content of the pulp, with reductions of up to 1.72% per unit increase in water salinity. In another study, Adriano et al. [37] obtained the respective values of 5.73 and 4.24 (g 100 g$^{-1}$) for ripe and semi-ripe West Indian cherry fruits.

For the phenolic compounds of West Indian cherry (Figure 7A), plants irrigated with water having an electrical conductivity of 4.0 dS m$^{-1}$ showed the highest values for this parameter under fertilization combinations $C_1$, $C_6$, and $C_{10}$, which differed ($p \leq 0.01$) from the other combinations. On the other hand, in the treatment with the water having lower electrical conductivity, combinations $C_3$, $C_7$, and $C_{10}$ showed the highest values of phenolic compounds, differing from the other fertilization combinations.

As observed in Figure 7A, the highest values of phenolic compounds in West Indian cherry in plants under salt stress are associated with fertilization combinations $C_1$, $C_4$, $C_5$, $C_6$, $C_8$, and $C_{10}$, which differ from the plants irrigated with the ECw de 0.6 dS m$^{-1}$ and fertilized with combinations $C_1$, $C_6$, and $C_8$. Therefore, the availability of the optimum amount of N in plants under conditions of high salt concentrations in irrigation water can increase $CO_2$ assimilation efficiency and other carbohydrate-related pathways, increasing the concentration of organic constituents in the fruit [46]. On the other hand, phosphorus acts in energy metabolism, favoring the synthesis of nucleic acids, coenzymes, and sugars [12] and increasing the concentration of phenolic compounds in West Indian cherry under adverse conditions caused by salt stress, thus improving pulp quality [9].

The flavonoid contents in the fruit pulp differ ($p \leq 0.01$) between the electrical conductivity levels (Figure 7B). Plants irrigated with ECw of 4.0 dS m$^{-1}$ and subjected to fertilization combinations $C_5$ and $C_{10}$ performed better ($p \leq 0.01$) than the other combinations (Figure 7B). On the other hand, in plants irrigated with the lower ECw (0.6 dS m$^{-1}$), the highest flavonoid content is associated with combination $C_1$, which differed ($p < 0.01$) from other fertilization combinations treatments (Figure 7B).

In the present study, the increase in the flavonoid content in pulp of West Indian cherry fruits under salt stress is mainly related to an adequate potassium supply, since the fertilization combination that most favored their production was $C_{10}$ (100-100-140% of the recommended N-P-K level). Potassium participates in some crucial functions in plants, e.g., the transport of amino acids and sugars to storage organs and enzyme activation [47]. Furthermore, the adequate supply of this macronutrient can increase its competition with other cations, especially Na$^+$ [48]. Therefore, these functions of potassium are essential in the cultivation of plants irrigated with water having high electrical conductivity levels.

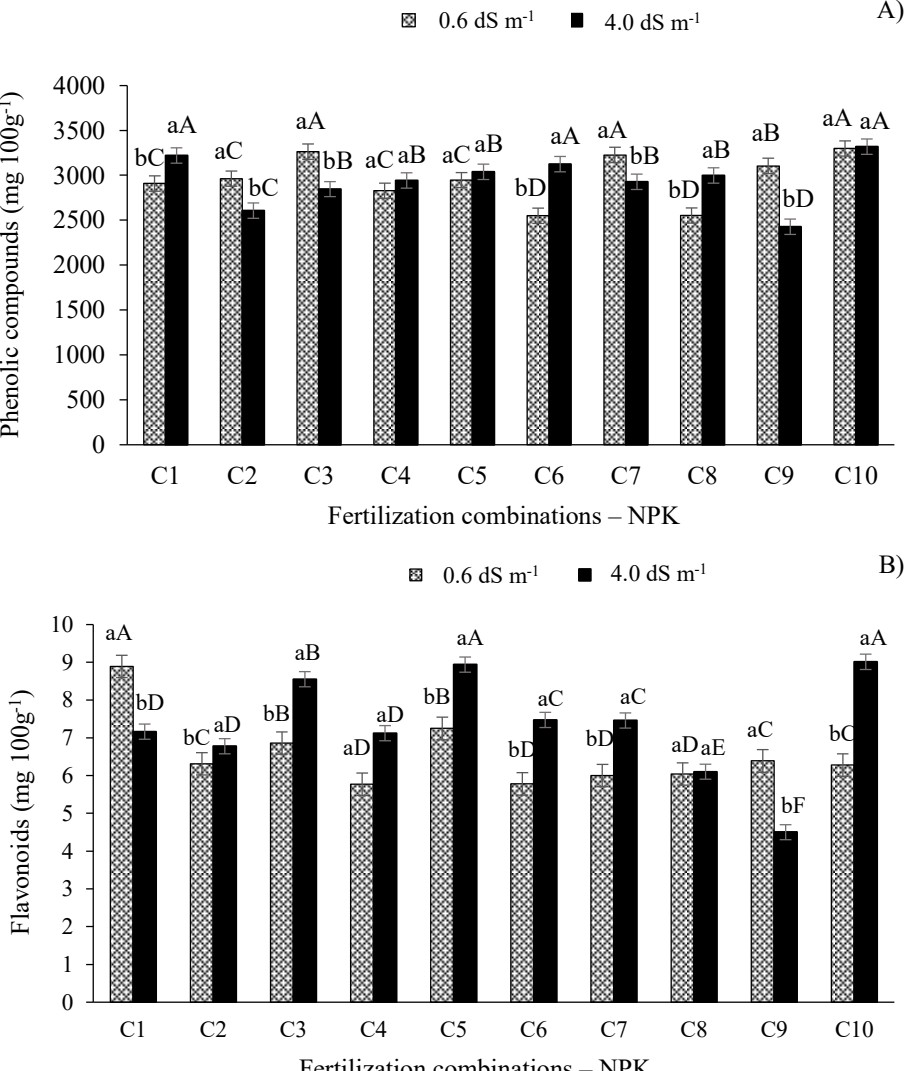

**Figure 7.** Phenolic compound (**A**) and flavonoids (**B**) in the pulp of West Indian cherry as a function of the interaction between the ECw and combinations of NPK fertilization in the second year of production. For description of fertilizer combinations (C) see Figure 2. Means followed by identical uppercase letters indicate that, for the same type of water, there are no significant differences between fertilization combinations by the Scott–Knott test at $p > 0.05$ probability. Identical lowercase letters in the same fertilization combination indicate no significant difference between salinity levels (F-test, $p > 0.05$). Mean of three replicates ± standard error.

According to the results shown in Figure 8, the content of anthocyanin in the pulp of West Indian cherry fruits were influenced ($p \leq 0.01$) by the electrical conductivity levels of irrigation water when paired with all fertilization combinations except $C_7$. In plants irrigated with ECw of 0.6 dS m$^{-1}$, the higher anthocyanin value was observed under combination $C_1$, differing from the other fertilization combinations. In plants irrigated with the ECw of 4.0 dS m$^{-1}$, the highest anthocyanin values were obtained under combinations $C_5$ and $C_{10}$, differing ($p \leq 0.01$) from the other combinations (Figure 8).

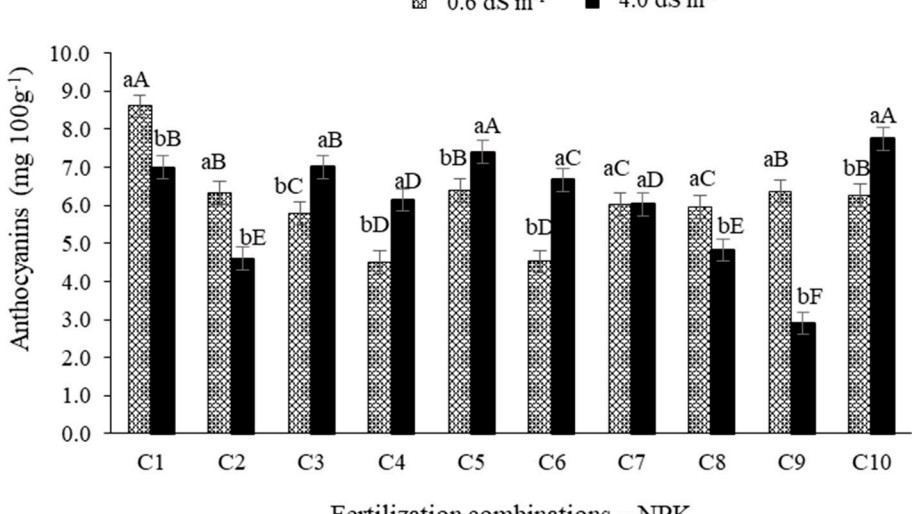

**Figure 8.** Anthocyanins (mg 100 $g^{-1}$) in the pulp of West Indian cherry fruits as a function of the interaction between the ECw and NPK fertilization combinations in the second year of production. For description of fertilizer combinations (C) see Figure 2. Means followed by identical uppercase letters indicate that, for the same type of water, there are no significant differences between fertilization combinations by the Scott–Knott test at $p > 0.05$ probability. Identical lowercase letters in the same fertilization combination indicate no significant difference between salinity levels (F-test, $p > 0.05$). Mean of three replicates ± standard error.

In the present study, the reduction in anthocyanin content of fruits observed in West Indian cherry plants irrigated with the higher ECw (4.0 dS $m^{-1}$) is possibly related to the reduction in the flavonoid content [49]. High anthocyanin values are essential for West Indian cherry since this compound is responsible for the red color of the ripe fruit, constituting one of the essential aspects regarding the interest of consumers [50]. Lacerda et al. [9] observed a reduction in anthocyanin content when salinity increased up to 4.3 dS $m^{-1}$ regardless of the fertilization combination with N-K.

From this perspective, it is necessary to adopt an increasingly effective management in crops subjected to salt stress, e.g., fertilization combinations with NPK.

## 4. Conclusions

The irrigation of West Indian cherry with water having a salinity of 4.0 dS $m^{-1}$ negatively affected all production variables, namely total number of fruits, total fruit weight, mean fruit weight, and equatorial fruit diameter, in the second year of production.

Plants irrigated with water at a salinity of 4.0 dS $m^{-1}$ and treated with the fertilization combination containing 100% N + 100% $P_2O_5$ + 120% $K_2O$ of recommended respective levels favored the total number of fruits and the total fruit weight of West Indian cherry in the second year of production.

The interaction between the electrical conductivity irrigation water of 0.6 dS $m^{-1}$ and the fertilization combination containing 100% N + 80% $P_2O_5$ + 120% $K_2O$ increased the total number of fruits and the total fruit weight of West Indian cherry in the second year of production.

West Indian cherry plants irrigated with water at an electrical conductivity of 4.0 dS $m^{-1}$ and paired with combinations $C_1$ (80-100-100%), $C_3$ (120-100-100%), $C_6$ (100-120-100%), and $C_{10}$ (100-100-140%) of the recommended N-P-K levels showed increases in titratable acidity, total soluble solids, reducing sugars, vitamin C, and phenolic compounds ($C_1$); TSS/TA ratio (maturity index), vitamin C and phenolic compounds ($C_3$); total soluble solids, reducing sugars, TSS/TA ratio (maturity index), phenolic compounds, flavonoids, and anthocyanins ($C_6$); vitamin C, TSS/TA ratio (maturity index), flavonoids, and anthocyanins ($C_{10}$) compared to plants irrigated with the electrical conductivity level of 0.6 dS $m^{-1}$.

West Indian cherry plants irrigated with water at an electrical conductivity of 4.0 dS m$^{-1}$ were negatively affected in terms of the TSS/TA ratio (maturity index), flavonoids, and anthocyanins ($C_1$); total soluble solids, reducing sugars, phenolic compounds, and flavonoids ($C_3$); vitamin C ($C_6$); titratable acidity; and phenolic compounds ($C_{10}$).

**Author Contributions:** Conceptualization, H.R.G., S.I.B. and L.T.S.A.; Data curation, H.R.G.; Formal analysis, A.M.d.S.F., A.A.R.d.S., S.I.B., L.T.S.A. and P.S.C.; Funding acquisition, H.R.G.; Investigation, A.M.d.S.F., H.R.G. and A.S.d.M.; Methodology, A.S.d.M., A.A.R.d.S. and R.L.d.S.F.; Project administration, H.R.G.; Resources, A.M.d.S.F., H.R.G. and A.S.d.M.; Software, A.M.d.S.F. and A.A.R.d.S.; Supervision, H.R.G. and A.S.d.M.; Validation, A.M.d.S.F., H.R.G., A.S.d.M., L.H.G.C. and R.M.F.d.F.; Visualization, A.A.R.d.S., R.L.d.S.F., L.H.G.C. and R.M.F.d.F.; Writing—original draft, A.M.d.S.F.; Writing—review & editing, A.M.d.S.F., H.R.G., A.S.d.M. and P.S.C. All authors have read and agreed to the published version of the manuscript.

**Funding:** This research was funded by National Council of Scientific and Technological Development (CNPq), reference number CNPq. 151309/2019-1.

**Institutional Review Board Statement:** Not applicable.

**Informed Consent Statement:** All authors have read revised manuscript including Acknowledgements and have given their consent.

**Data Availability Statement:** Data are contained within the article. No supplemental data is provided.

**Acknowledgments:** The authors thank the Postgraduate Program in Agricultural Engineering of the Federal University of Campina Grande, the State University of Paraíba, the National Council of Scientific and Technological Development (CNPq), and the Coordination for the Improvement of Higher Education Personnel (CAPES) for their support for this research. Thanks to Fellow Leandro de Padua Souza, who implemented and conducted experiment in the first year.

**Conflicts of Interest:** The authors declare no conflict of interest. The funders had no role in the design of the study; in the collection, analyses, or interpretation of data; in the writing of the manuscript; or in the decision to publish the results.

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
