# Peer review of "Production and Quality of West Indian Cherry (Malpighia emarginata D. C.) under Salt Stress and NPK Combinations"

_horticulturae, doi:10.3390/horticulturae9060649_

Round 1

Reviewer 1 Report

Reviewer's opinion

The manuscript basically meets the requirements, it’s topic is very relevant for the practice. However, I would like to make some comments in order to improve this work:

Title:

The Latin name of the plant species should also be given here.

Abstract:

A lot of important information are missing here, e.g. the chemical characteristics of fruit pulp, these should also be included (what properties have been tested and how they have been affected by the different treatments).

Keywords:

The main examined properties should also be listed here.

Materials and Methods:

How old plants were examined during the experiment?

Two varieties were also mentioned as plant materials (cultivars Junco and Flor Branca). Which variety was actually tested? There are no further data about cultivars in the paper, so presumably only one of the cultivars was used as plant material (if both, that is a problem).

It would have been very good if in the experiment a ’real control’ had been used too: a unit receiving the C2 NPK dose but irrigated with normal, salt-free water.

The data in Table 1 should also be evaluated in one or two sentences: Was the soil well supplied with nutrients? Had it compact or loose texture? etc.

Results and discussion:

This sub-chapter is too long. The evaluation of yield (basically 3 characteristics) is almost 5 pages long, while the analysis of nutritional values (9 characteristics) is 7 pages long.

The figures do not show the exact data. Either the actual values should be indicated on or below the columns or they should be presented in tabular form (instead of figures). If data were presented in one or two larger tables, space could be saved (there are not so many titles, the same multi-line explanations/legends everywhere).

For figures, the legend is usually given below the figure’s title. For tables, it is given below the tables.

Don’t put punctuation marks at the end of the titles of figures and tables!

The evaluation should be a bit more standardised for the traits under consideration (also for TNF, TFW, MFW, where the data are perhaps a bit over-explained). Where similar trends are found, the traits could be evaluated together.

In case of TSS/TA, what is the preferred ratio among consumers?

During the evaluation, irrigation water of 4.0 dS m-1 is consistently called "highest" and 0.6 dS m-1 as "lowest". However, as these are the only two treatments, the designation "higher" and "lower" would be better.

As for the chemical characteristics of fruit pulp, the effects of treatments are not very unambiguous, neither in terms of salinity of the irrigation water nor in terms of NPK supply. Thus, you must be cautious in drawing conclusions. But some conclusion is necessary, whether that further investigations are needed to clarify the issue.

Conclusions:

You should also write about chemical characteristics of fruit pulp here, since more than half of the manuscript is about them, even if the results are contradictory. The effects of irrigation with 0.6 dS m-1 water should also be summarised, not just irrigation with ECw of 4.0 dS m-1.

The treatments highlighted in paragraph 4 within conclusion chapter are not very similar, it is not sure that these should be highlighted…

References:

The first reference is not formally correct.

In general:

There are some stylistic and typing errors in the text which need to be corrected.

Author Response

Campina Grande, PB, Brszil.

May 25, 2023

Manuscript ID: Horticulturae-2423105

Dear Editor and Reviewer

The authors are very grateful to you and the reviewers for the positive and constructive comments and suggestions on our manuscript titled “Production and quality of West Indian cherry (Malpighia emarginata D. C.) under salt stress and NPK combinations”. The authors inform that a meticulous revision of the manuscript was made, incorporating the suggestions and adopting the text according to the comments. Attached is the revised version of the manuscript. All changes in text are highlighted in red color.

The authors remain at your disposal for any additional information and clarification.

The answers/clarifications to the issues raised by the Reviewers/Editor are presented below:

REVIEWER 1

  1. Title: The Latin name of the plant species should also be given here.

Response: The Latin name of the plant species was added and new title is “and new title is “Production and quality of West Indian cherry (Malpighia emarginata D. C.) under salt stress and NPK fertilization combinations”.

  1. Abstract: A lot of important information are missing here, e.g. the chemical characteristics of fruit pulp, these should also be included (what properties have been tested and how they have been affected by the different treatments).

Response: The following text has been added: Production and post-harvest variablesevaluated were: the total fruit weight, total number of fruits, mean fruit weight, the polar and equatorial diameter and total soluble solids, pulp pH, titratable acidity, maturity ratio, vitamin C,  reducing sugars, total phenolic compounds,  total anthocyanins, and flavonoids, respectively.

  1. Keywords: The main examined properties should also be listed here.

Response: new keywords Post-harvest; flavonoids; anthocyanins; added              

  1. Materials and Methods:

  • How old plants were examined during the experiment?

Response: In this article, the plants were evaluated in the second year of production (after 375 days of planting), starting production with one year and nine months old, ending with two years old. Information added to the text.

  • Two varieties were also mentioned as plant materials (cultivars Junco and Flor Branca). Which variety was actually tested?

Response: There are no further data about cultivars in the paper, so resumably only one of the cultivars was used as plant material (if both, that is a problem).

Response: Since it is a grafted plant, with a graft of Flor Branca and a rootstock of Junco (information provided in the first draft of manuscript page 3, second paragraph).

  • It would have been very good if in the experiment a ’real control’ had been used too: a unit receiving the C2NPK dose but irrigated with normal, salt-free water.

Response: In fact, a salinity of 0.6 dS m-1 was used as a control ( alow salinity water), as it is below the threshold salinity of the West Indian cherry (1.16 dS m-1).

  • The data in Table 1 should also be evaluated in one or two sentences: Was the soil well supplied with nutrients? Had it compact or loose texture? etc.

Response: The following text has been added to the final version of the manuscript:The soil presents sandy clay loam texture, adequate levels of P, K, Ca, Mg, and satisfactory pH and organic matter. At planting, the soil was supplied with 20 g P2O5 and K2O as basal dose.

  1. Results and Discussion:

  • This sub-chapter is too long. The evaluation of yield (basically 3 characteristics) is almost 5 pages long, while the analysis of nutritional values (9 characteristics) is 7 pages long.

Response: The authors consider it important to present and discuss in detail the production data because most of the time farmers made their decisions with regard to production.

  • The figures do not show the exact data. Either the actual values should be indicated on or below the columns or they should be presented in tabular form (instead of figures). If data were presented in one or two larger tables, space could be saved (there are not so many titles, the same multi-line explanations/legends everywhere).

Response: As there is a lot of information, the authors chose to present the data in graphs, as they believe it is easy to understand. The authors chose a compatible scale of Y-axis so that column heigt gives an idea of data. In the opinion of authors, putting numbers on the column, seems a non-viable alternative because of size of font to be used.

  • For figures, the legend is usually given below the figure’s title. For tables, it is given below the tables.

Response: The subtitles for the titles of tables and figures were prepared in accordance with the norms of Horticulturae.

  • Don’t put punctuation marks at the end of the titles of figures and tables!

Response: The subtitles for the titles of tables and figures were prepared in accordance with the norms of Horticulturae, and use the Microsoft Word template provided by publisher.

  • The evaluation should be a bit more standardised for the traits under consideration (also for TNF, TFW, MFW, where the data are perhaps a bit over-explained). Where similar trends are found, the traits could be evaluated together.

Response: The authors understand that traits could be evaluated together; however, for the reasons given earlier and better understanding, the authors opted otherwise.

  • In case of TSS/TA, what is the preferred ratio among consumers?

Response: The following text was added to the final version of the manuscript: while the optimum value is 6.8 [40].

  • During the evaluation, irrigation water of 4.0 dS m-1is consistently called "highest" and 0.6 dS m-1 as "lowest". However, as these are the only two treatments, the designation "higher" and "lower" would be better.

  Response: Authors are thankful for correcting this mistake. The terms "higher" and "lower" have been used in the final version of the manuscript in place of highest and lowest.

  • As for the chemical characteristicsof fruit pulp, the effects of treatments are not very unambiguous, neither in terms of salinity of the irrigation water nor in terms of NPK supply. Thus, you must be cautious in drawing conclusions. But some conclusion is necessary, whether that further investigations are needed to clarify the issue.

Response: The authors agree that more investigations are needed to clarify the issue, in which future studies will be carried out. It is emphasised at the end of conclusions.

  1. Conclusions:

  • You should also write about chemical characteristicsof fruit pulp here, since more than half of the manuscript is about them, even if the results are contradictory. The effects of irrigation with 0.6 dS m-1 water should also be summarised, not just irrigation with ECw of 4.0 dS m-1.

Response: New conclusions have been added including the effects of irrigation with water of 0.6 dS m-1.

  • The treatments highlighted in paragraph 4 within conclusion chapter are not very similar, it is not sure that these should be highlighted…

  Response: Text modified and new conclusions have been added.

  1. References:

Response: The reference has been adjusted.

Reviewer 2 Report

Dear authors

The article entitled Production and quality of West Indian cherry under salt stress

and NPK fertilization combinations, in general, seems well written, here you will find my comments to improve it.

Include the experimental working hypothesis in the introduction.

Figure 1 corresponds to the results obtained, place in the corresponding section.

Table 1 are results, place in the corresponding section.

In the results, significant differences are mentioned as p≤0,01 (it should say p≤0.01), review the entire text and correct.

In Table 3, standardize the nomenclature for all non-significant data (** and ns).

Homogenize the significant differences in the whole text as p≤0.05 (if it gives less, it does not matter, it is denoted that it is significant).

Perform a comprehensive analysis of the results for polyphenols, anthocyanins, and flavonoids (two of them make up a fraction of another).

The conclusions respond to the proposed objectives and the proposed hypothesis is contrasted.

Minor editing of English language required

Author Response

Campina Grande, PB, Brszil.

May 25, 2023

Manuscript ID: Horticulturae-2423105

Dear Editor and Reviewer

The authors are very grateful to you and the reviewers for the positive and constructive comments and suggestions on our manuscript titled “Production and quality of West Indian cherry (Malpighia emarginata D. C.) under salt stress and NPK combinations”. The authors inform that a meticulous revision of the manuscript was made, incorporating the suggestions and adopting the text according to the comments. Attached is the revised version of the manuscript. All changes in text are highlighted in red color.

The authors remain at your disposal for any additional information and clarification.

The answers/clarifications to the issues raised by the Reviewers/Editor are presented 

REVIEWER 2

  1. Include the experimental working hypothesis in the introduction.

Response: Hypothesis has been added in the revised version of the manuscript in the penultimate paragraph of Introduction.

  1. Figure 1 corresponds to the results obtained, place in the corresponding section.

Response: The Figure 1 is not discussed in the results. In the methodology, it only serves to inform the readers of the temperature and relative air humidity conditions during the conduction of the experiment.

  1. Table 1 are results, place in the corresponding section.

Response: The Table 1 is not discussed in the results. In the methodology, it only serves to inform the readers of the chemical and physical characteristics of the soilbefore  the conduction of the experiment.

  1. In the results, significant differences are mentioned as p≤0,01 (it should say p≤0.01), review the entire text and correct.

Response: Authors are thankful to reviewer for pointing out this mistake. The text has been corrected in the revised version of the manuscript.

  1. In Table 3, standardize the nomenclature for all non-significant data (** and ns).

Response: The text was corrected in the revised version of the manuscript.

  1. Homogenize the significant differences in the whole text as p≤0.05 (if it gives less, it does not matter, it is denoted that it is significant).

Response: The text was corrected in the revised version of the manuscript. However, when it was not significant p>0.05 was maintained.

  1. Perform a comprehensive analysis of the results for polyphenols, anthocyanins, and flavonoids (two of them make up a fraction of another).

Response: As suggested the text was revised in the new draft of the manuscript.

Reviewer 3 Report

The current study assessed the effects of water salinity and different NPK fertilization rates on the quantity and quality of West Indian cherry. The authors conducted a detailed analysis, comparing various variables, and revealed the positive influence of potassium and phosphorus on fruit production and quality, respectively, under salt stress. These findings are crucial for local farmers to adjust the NPK fertilizer ratio based on the specific soil and water conditions in their region. The manuscript is well-written, and the results are effectively presented.

One major concern I have pertains to whether the authors ranked the C1 to C9 fertilization ratios based on their impact on both fruit production and quality under each irrigation condition. For instance, the authors could utilize membership functions to identify the optimal fertilization ratio for regions with lower-salinity irrigation water. Such results would greatly assist others in selecting an appropriate strategy. Otherwise, individuals may face a dilemma in compromising between quantity and quality.

Page 13. Please delete C8 in the sentence of “On the other hand, the plants irrigated with the ECw of 0.6 dS m-1 showed higher values of vitamin C when they plants received combinations C2, C6, C7 C8, and C9”.

Page 15. Please verify the sentence “C6, C8, and C10, which differ from the plants irrigated with the ECw de 0.6 dS m-1 under”.

Page 16. I cannot understand the sentence of “In the present study, the reduction in the anthocyanin contents of fruits observed in West Indian cherry plants irrigated with the highest ECw (4.0 dS m-1) is possibly related to the reduction in the flavonoid contents [47].”

Author Response

Campina Grande, PB, Brszil.

May 25, 2023

Manuscript ID: Horticulturae-2423105

Dear Editor and Reviewer

The authors are very grateful to you and the reviewers for the positive and constructive comments and suggestions on our manuscript titled “Production and quality of West Indian cherry (Malpighia emarginata D. C.) under salt stress and NPK combinations”. The authors inform that a meticulous revision of the manuscript was made, incorporating the suggestions and adopting the text according to the comments. Attached is the revised version of the manuscript. All changes in text are highlighted in red color.

The authors remain at your disposal for any additional information and clarification.

The answers/clarifications to the issues raised by the Reviewers/Editor are presented 

 REVIEWER 3

  1. One major concern I have pertains to whether the authors ranked the C1 to C9 fertilization ratios based on their impact on both fruit production and quality under each irrigation condition. For instance, the authors could utilize membership functions to identify the optimal fertilization ratio for regions with lower-salinity irrigation water. Such results would greatly assist others in selecting an appropriate strategy. Otherwise, individuals may face a dilemma in compromising between quantity and quality.

Response. The authors consider this idea to be very important and that, at the moment, it is not possible to incorporate it into the article. However, in future studies, the authors could use membership functions to identify the optimal fertilization ratio for regions with lower-salinity irrigation water.

  1. Page 13. Please delete C8 in the sentence of “On the other hand, the plants irrigated with the ECw of 0.6 dS m-1 showed higher values of vitamin C when they plants received combinations C2, C6, C7 C8, and C9”.

Response: The text was excluded in the revised version of the manuscript, as suggested.

  1. Page 15. Please verify the sentence “C6, C8, and C10, which differ from the plants irrigated with the ECw de 0.6 dS m-1 under”.

Response: The sentence was checked and modified adding:when compared to ECw of 4.0 dS m-1.

  1. Page 16. I cannot understand the sentence of “In the present study, the reduction in the anthocyanin contents of fruits observed in West Indian cherry plants irrigated with the highest ECw (4.0 dS m-1) is possibly related to the reduction in the flavonoid contents [47].”

Response: The reduction in the anthocyanin contents is possibly related to the reduction in the flavonoid contents, because anthocyanins are flavonoids that are widely distributed in nature.

On behalf of the authors, I thank once again the unanimous Reviewers and Editor for suggestions and comments on our paper.

Antônio Manoel
